# Flow Characteristics by Blood Speckle Imaging in Non-Stenotic Congenital Aortic Root Disease Surrounding Valve-Preserving Operations

**DOI:** 10.3390/bioengineering12070776

**Published:** 2025-07-17

**Authors:** Shihao Liu, Justin T. Tretter, Lama Dakik, Hani K. Najm, Debkalpa Goswami, Jennifer K. Ryan, Elias Sundström

**Affiliations:** 1Department of Mathematics, KTH Royal Institute of Technology, 114 28 Stockholm, Sweden; shihaoli@kth.se (S.L.); jryan@kth.se (J.K.R.); 2Congenital Valve Procedural Planning Program, Department of Pediatric Cardiology and Division of Pediatric Cardiac Surgery, The Heart, Vascular, and Thoracic Institute, Cleveland Clinic, Cleveland, OH 44195, USA; trettej3@ccf.org (J.T.T.); dakikl2@ccf.org (L.D.); 3Department of Pediatric Cardiology, Cleveland Clinic, Cleveland, OH 44195, USA; najmh@ccf.org; 4Cardiovascular Innovation Research Center, Department of Cardiovascular Medicine, The Heart, Vascular and Thoracic Institute, Cleveland Clinic, Cleveland, OH 44195, USA; 5Department of Engineering Mechanics, FLOW Research Center, KTH Royal Institute of Technology, 100 44 Stockholm, Sweden

**Keywords:** blood speckle imaging, congenital heart disease, aortic valve repair, hemodynamics, aortic valve-sparing root replacement, aortic regurgitation

## Abstract

Contemporary evaluation and surgical approaches in congenital aortic valve disease have yielded limited success. The ability to evaluate and understand detailed flow characteristics surrounding surgical repair may be beneficial. This study explores the feasibility and utility of echocardiographic-based blood speckle imaging (BSI) in assessing pre- and post-operative flow characteristics in those with non-stenotic congenital aortic root disease undergoing aortic valve repair or valve-sparing root replacement (VSRR) surgery. Transesophageal echocardiogram was performed during the pre-operative and post-operative assessment surrounding aortic surgery for ten patients with non-stenotic congenital aortic root disease. BSI, utilizing block-matching algorithms, enabled detailed visualization and quantification of flow parameters from the echocardiographic data. Post-operative BSI unveiled enhanced hemodynamic patterns, characterized by quantified changes suggestive of the absence of stenosis and no more than trivial regurgitation. Rectification of an asymmetric jet and the reversal of flow on the posterior aspect of the ascending aorta resulted in a reduced oscillatory shear index (OSI) of 0.0543±0.0207 (pre-op) vs. 0.0275±0.0159 (post-op) and p=0.0044, increased peak wall shear stress of 1.9423±0.6974 (pre-op) vs. 3.6956±1.4934 (post-op) and p=0.0035, and increased time-averaged wall shear stress of 0.6885±0.8004 (pre-op) vs. 0.8312±0.303 (post-op) and p=0.23. This correction potentially attenuates cellular alterations within the endothelium. This study demonstrates that children and young adults with non-stenotic congenital aortic root disease undergoing valve-preserving operations experience significant improvements in flow dynamics within the left ventricular outflow tract and aortic root, accompanied by a reduction in OSI. These hemodynamic enhancements extend beyond the conventional echocardiographic assessments, offering immediate and valuable insights into the efficacy of surgical interventions.

## 1. Introduction

Aortic valve repair or preservation has emerged as a preferred option for addressing congenital aortic root pathologies in children and young adults. However, achieving successful outcomes hinges upon a comprehensive grasp of the intricate anatomy of the congenitally malformed aortic root and its valve, and the understanding and technical proficiency to restore normal valve geometry and mechanics [1]. Accurately assessing the repaired valve holds paramount importance in appraising immediate surgical outcomes and optimizing post-operative care. While traditional echocardiographic techniques offer valuable insights, they may falter in capturing nuanced hemodynamic characteristics, which may enhance understanding [2].

Continuous-wave Doppler echocardiography stands as a cornerstone in clinical practice for assessing the severity of aortic stenosis. This method entails the measurement of peak blood flow velocity traversing the aortic valve during systole. Utilizing the modified Bernoulli equation, clinicians derive an estimation of the transvalvar pressure gradient [3]. This modality holds preference over cardiac catheterization owing to its accessibility, cost-effectiveness, and non-invasive means, thereby enhancing its widespread utilization in clinical settings [4].

Continuous-wave Doppler echocardiography is primarily confined to estimating peak instantaneous velocity and transvalvar pressure gradient, a limitation when compared to equations governing the complete hemodynamic profile at the site of greatest constriction [5,6]. Peak velocity measurements disregards the momentum inherent in blood flow across the entire vascular cross-section, a critical factor in accurately assessing velocity profiles and wall shear stress (WSS). Furthermore, peak velocity estimation via Doppler echocardiography depends on the angle of insonation being parallel to the direction of flow. Minimal misalignment may significantly underestimate the interrogated peak velocity [7]. While various non-invasive alternatives have been explored, their integration into clinical practice remains pending [6,8].

Blood speckle imaging (BSI) has emerged as a novel non-invasive mean to visualize and quantify blood flow characteristics [2,9,10]. This modality entails direct measurement and visualization of both the magnitude and direction of the blood flow field, sampled at an ultra-high (kilohertz) frequency range [9,11,12]. BSI presents a promising modality for overcoming the limitations associated with conventional Doppler echocardiography, including angle dependence and reliance on single peak velocity acquisitions [5]. In addition, BSI allows the calculation of WSS based on velocity information within a cross-sectional profile. BSI has been used in recent studies to assess aortic valve flow dynamics. Carnotti et al. (2023) [13] demonstrate the feasibility and reproducibility of BSI in evaluating aortic flow patterns in healthy children. The authors quantified vortex formation in the aortic root and ascending aorta, providing baseline data that can be used for comparison in pathological conditions. In a pilot study by Park et al. (2023) [14], the authors investigates the relationship between trans-aortic turbulence, as measured by BSI, and aortic valve inflammation, assessed using 18F-NaF PET/CT. The findings suggest that higher degrees of turbulence are associated with increased inflammatory activity in the aortic valve, highlighting the potential of BSI in evaluating disease progression in aortic stenosis.

Our group has previously explored the biomechanics of bicuspid aortic valves (BAVs) through fluid–structure interaction (FSI) modeling [15,16]. These computational studies provided detailed insight into the mechanical stresses and flow separation regions in BAVs with variation in the interleaflet triangle and commissural angle. However, numerical modeling lacks immediate clinical applicability in perioperative settings. The current BSI study builds upon and complements our prior FSI work by offering in vivo, patient-specific hemodynamic measurements. BSI enables bedside assessment of WSS and OSI, which were previously inferred via simulation. This work thus bridges the gap between computational and clinical hemodynamic characterization.

A recent study has evaluated BAV-specific hemodynamics using imaging and modeling techniques. Anam et al. (2025) [17] used patient-specific computational modeling to evaluate and compare post-TAVR complications, specifically paravalvular leak (PVL) and thrombogenic risk, in BAV patients implanted with older versus newer generation TAVR devices, demonstrating improved outcomes with newer devices.

A perioperative echocardiography study by Hayaschi et al. (2021) showed that patients undergoing VSRR surgery had a reduced oscillatory shear index (OSI) and increased peak WSS compared to controls, suggesting a reduced leaflet stress and potential degeneration following VSARR [18]. A recent proteomic study in LVAD recipients linked abnormal WSS/OSI patterns to the reduced expression of cytoskeletal and junction proteins in aortic leaflets, implicating shear forces in endothelial dysfunction, inflammation, and tissue remodeling [19]. While these findings support a mechanical link between WSS/OSI and structural valve changes, most evidence is based on limited longitudinal data correlating shear stress to clinical outcomes.

This retrospective, small cohort study aims to assess the feasibility and application of BSI for pre- and post-operative transesophageal echocardiographic (TEE) evaluation of flow characteristics and resulting tissue biomechanics of the aortic valve leaflets and adjacent arterial walls in children and adults with congenital aortic valve disease undergoing valve repair and valve-sparing aortic root replacement.

## 2. Methods

### 2.1. Surgical Cohort

A retrospective analysis of a single-center, cohort study was conducted. Ten patients with congenital (neo-)aortic valve disease and/or (neo-)aortic root dilation who met clinical criteria for aortic valve repair or valve-sparing root replacement were included. Intraoperative TEE was performed as part of the standard pre- and post-operative evaluation surrounding the surgical procedure. For analysis, patients where divided into two subcohorts: those with a well-functioning aortic valve versus those with valve dysfunction. Aortic valve dysfunction was defined as greater than mild stenosis and/or greater than mild regurgitation.

### 2.2. Quantification of the Velocity Field

BSI analyzes sequences of ultrasound images by tracking the movement of small “speckle” patterns made by moving red blood cells. This tracking is performed using a block-matching algorithm that searches for the “best match” between successive frames. Further details are available in [9,10]. BSI acquisition was obtained using the Vivid E95 ultrasound system (GE HealthCare, GE Vingmed Ultrasound AS, Horten, Norway). The acquired data consisted of a cine loop comprising standard 2D sonographic images, blood velocity data, a corresponding confidence level map, and associated metadata. The tissue data formed a single channel, where pixel intensity directly corresponded to image brightness.

Blood velocity data were multichannel and subsequently processed into a velocity field. A separate single channel provided quantitative information on the confidence level associated with the blood velocity estimates. This confidence level ranged from 0 (low confidence) to 1 (high confidence) and reflected the fidelity of the block-matching algorithm used to track blood speckle displacement. Lower confidence values indicated potentially unreliable velocity estimates, often caused by factors such as poor image quality or out-of-plane blood speckle movement [20,21,22].

The spatial resolution of the BSI images was 0.0825 mm/pixel. The dimensions of the acquired tissue data in the x- and y-directions provided the total number of pixels. This information was then used to generate a 2D grid encompassing the region of interest (ROI). Subsequently, the velocity field was mapped onto this 2D grid, facilitating both visualization and quantification of blood flow patterns. The BSI images were acquired with a pulse repetition frequency (PRF) of 6 kHz. The velocity encoding setting allowed for a maximum trackable blood velocity of up to 2 m/s, which was sufficient for both pre- and post-operative evaluations.

### 2.3. Hemodynamic Characterization Using Wall Shear Stress Related Indicators

Endothelial cells lining the aortic wall are sensitive to the mechanical forces exerted by blood flow. To evaluate these forces and their potential impact on endothelial cell health, this study employed three key hemodynamic parameters: peak wall shear stress (PeakWSS), time-averaged wall shear stress (TAWSS), and oscillatory shear index (OSI).

Wall shear stress (WSS) represents the frictional force exerted by blood flow per unit area on the aortic wall. It varies spatially and temporally depending on blood flow patterns and the geometry of the thoracic aorta and aortic valve. PeakWSS refers to the maximum shear stress experienced at the wall of a blood vessel at peak systole and has been correlated with disease progression and assessing the risk of aneurysm rupture. Time-averaged wall shear stress (TAWSS) reflects the localized average WSS over the cardiac cycle [23,24,25].(1)$$TAWSS=\frac{1}{T}\int_{0}^{T}|WSS_{i}|dt.$$

While TAWSS provides a magnitude estimate, spatial variations in WSS offer insights into the non-uniformity of the forces experienced by the endothelium. However, these variations do not capture the frequency of flow direction changes. Therefore, OSI quantifies the oscillatory nature of WSS. OSI ranges from 0 (perfectly steady flow) to 0.5 (maximal flow reversal [24].(2)$$OSI=\frac{1}{2}\left(1-\frac{|\int_{0}^{T}WSS_{i}dt|}{\int_{0}^{T}|WSS_{i}|dt}\right).$$

### 2.4. Statistical Analysis

The study aimed to assess the statistical significance of differences in PeakWSS, TAWSS, and OSI, before and after surgery. Normality of the parameter differences was evaluated using the Shapiro–Wilk test. For normally distributed data, a one-sample Student’s *t*-test was employed. With a sample size (*n*) of 10, the mean difference ($\Delta \bar{x}$) and standard deviation (SD) were calculated. The resulting statistic,(3)$$t=\frac{\Delta \bar{x}-\mu_{0}}{SD/\sqrt{n}}.$$ Consequently, a rejection of the null hypothesis, that the surgery does not have an effect on the hemodynamic markers, is given by the probability of this effect occurring by chance was calculated as:(4)$$2\int_{-\infty }^{t}\rho_{t}\left(x\right)dx$$
which denotes the cumulative distribution function of the Student *t*-distribution. A significance level of 0.05 was chosen for hypothesis testing. The findings were graphically represented using box-and-whisker plots to illustrate changes in the hemodynamic markers induced by surgery.

## 3. Results

A total of 10 patients were included in the study. Patient demographics, body size, cardiac diagnoses, surgical indications, aortic valve morphology, and surgery performed are presented in Table 1. Pre-operative dimensions and pre- and post-operative echocardiographic assessment of the aortic valve function are presented in Table 2. The mean pre- and post-operative aortic valve virtual basal ring, or “annulus” diameters, were 30.2±5.5 mm and 17.1±3.2 mm (p<0.001), respectively. All patients had some degree of aortic root dilation, with all but one patient (Patient 10) having some degree of aortic regurgitation and/or stenosis. All patients in Group 1, with significant aortic regurgitation, showed mild or no aortic regurgitation following surgery. All patients in Group 2, with well-functioning aortic valves, remained well-functioning post-operatively without significant stenosis or regurgitation.

There was no significant difference in the entire cohort in the mean of the peak systolic aortic valve velocity pre- and post-surgery (1.1584±0.3919 m/s vs. 1.1290±0.2144 m/s, p=0.8995, Figure 1). Post-surgery reveals a notable reduction in the standard deviation of the peak velocity between peak systole and towards valve closure. This indicates a more uniform and stabilized flow profile following the surgical intervention.

### 3.1. Qualitative Flow Field Evaluation of BSI

Figure 2 shows the flow through a transesophageal long-axis view of the left ventricular outflow tract and aortic valve during different stages of the cardiac cycle for both the pre- and post-operative cases of Patient 1. The flow pattern differed pre-operatively from that post-operatively (Figure 2). Pre-operatively, Patient 1 exhibited a dilated aortic root and ascending aorta showing eccentric and less organized flow pre-surgery. This demonstrated an uneven jet pattern across the aortic valve in systole. Post-surgery, with adjustments toward normal valvar morphology and reduced aortic virtual basal ring diameter, the flow became more laminar and unidirectional. This behavior was consistent in 9 out of 10 patients.

### 3.2. Velocity Profile and Shear Stress Quantification

The blood flow illustrated in Figure 2 is further analyzed along the vertical profile (white dashed line) for Patient 1, see Figure 3. In the pre-operative scenario, the long-axis velocity component is initially low during early systole but increases towards peak systole, forming an anteriorly directed jet with flow reversal on the posterior side of the aortic root. This results in a significant velocity gradient at the jet’s shear layer [26], leading to increased shear stress. By post-systole, the velocity magnitude decreases, and the jet’s center shifts approximately 5 mm in the short-axis direction.

In the post-operative scenario, a relatively symmetric top-hat velocity profile emerges at peak systole, with minimal recirculation near the endothelial wall. Here, the shear layer on the anterior aortic wall exhibits a steeper gradient, resulting in higher shear stress levels. Additionally, there is less displacement of the jet’s center along the short axis compared to the pre-operative case.

Figure 4 complements Figure 3 by illustrating the statistical mean and standard deviation for Patients 1–10 at peak systole. Pre-operatively, the impaired flow is evident towards the right/left coronary leaflets, with the jet directed more anteriorly, c.f., Figure 3a,b. Post-operatively, the statistics indicate a fuller velocity profile at peak systole, with larger velocities directed anteriorly towards the probe. The greater asymmetry observed pre-operatively causes a reduced velocity gradient on this side compared to the post-operative case, resulting in lower axial wall shear stress and indicating a tendency for retrograde flow; see Figure 4b. On the opposite side (y/Y=1), i.e., the posterior aortic wall, the gradient is similarly small with low axial velocity, with some patients indicating changed sign of the wall shear stress, implying flow reversal pre-operatively as compared to the post-operative case.

### 3.3. Quantification of PeakWSS, TAWSS, and OSI Parameters

Statistics of the PeakWSS, TAWSS, and OSI parameters are quantified in Table 3 for Patients 1–10 in both pre- and post-operative cases.

The difference in these parameters is given in quantile–quantile (Q-Q) plots in Figure 5 and shows that the data falls on a straight line, indicating normally distributed data.

The statistics PeakWSS, TAWSS, and OSI are also presented with box-and-whisker plots in Figure 6. PeakWSS demonstrates a significant increase from a pre-operative value of 1.94 Pa to a post-operative value of 3.69 Pa (*p*=0.0035), indicating a marked change in wall shear dynamics following intervention. This increase is attributed to a steeper velocity gradient near the anterior wall of the aorta (y/Y=0), specifically closer to the imaging probe, as illustrated in Figure 3. Post-operatively, the flow becomes more directed and coherent, resulting in higher near-wall velocities and thus a greater shear stress at the vessel boundary. It is also evident in Figure 6a that the change in PeakWSS is more pronounced for Group 1 compared to Group 2 in the cohort. This is also evidenced in Table 3. In both pre-operative and post-operative cases, the axial velocity undergoes a sign change, transitioning from positive to negative flow, coinciding with the location of the shear layer. Pre-operatively, the retrograde flow towards the posterior wall is stronger than post-operatively. Integrating over the cardiac cycle reveals that the mean value of the TAWSS increases from 0.6885 Pa to 0.8312 Pa, with a larger standard deviation, as shown in Figure 6. When WSS is integrated over the entire cardiac cycle, the influence of transient flow features—such as retrograde flow—is effectively averaged out. As a result, the difference in TAWSS between pre- and post-operative states becomes statistically insignificant (p=0.23). However, retrograde flow with a change in flow direction results in a non-zero OSI in the axial flow direction. The mean values for OSI significantly decreases in the mean from 0.0534 to 0.0275. Thus, Figure 6 indicates that the OSI is significantly larger pre-operatively than post-operatively, which is statistically significant (p=0.0044). Periods of significant wall shear stress combined with a non-zero OSI have been suggested as a risk factor for cell-driven changes in the aortic wall structure, including the development and progression of atherosclerosis.

OSI was calculated using Equation (Equation 2). For each patient, the wall region (anterior or posterior) that contributed most significantly to the OSI, as defined by Equation (Equation 2), was identified. The OSI values reported in Table 3 were calculated using data from this dominant wall region. For the majority of patients, the anterior wall (y/Y=0) exhibited the highest OSI contribution. However, for Patient 4, the posterior wall (y/Y=1) showed the greatest contribution, and thus this region was used in the OSI calculation for that patient.

### 3.4. Relationship with Aortic Virtual Basal Ring Diameter

The correlation between aortic virtual basal ring diameter and OSI is weak and positive ($R^{2}=0.187,r=0.3755,\;\mathrm{and}\;p=0.0571$); however, it does not correlate to the same extent with TAWSS ($R^{2}=0.0192,r=0,\;\mathrm{and}\;p=0.56$). Figure 7a shows that the relationship between PeakWSS and the virtual basal ring diameter is almost significant ($R^{2}$ = 0.304, *r* = 0.5148, and *p* = 0.0118), indicating a weak negative correlation.

## 4. Discussion

In this study we utilized BSI for evaluating aortic valve repair or valve-sparing root replacement (VSRR) in a cohort of 10 patients, leveraging the capabilities of the newly available mini-3D transesophageal probe. The findings indicated that pre-operative BSI measurements exhibited unorganized eccentric flow, while post-operative BSI measurements showed uniform and laminar blood flow.

PeakWSS shows a significant increase from pre-operatively to post-operatively (p=0.0035). This rise is attributed to a steeper velocity gradient near the anterior wall of the aorta, closer to the imaging probe, as illustrated in Figure 3. The post-operative flow pattern results in higher near-wall velocities in this region, leading to elevated shear stress. The significant rise in PeakWSS suggests a biomechanical shift in local hemodynamics, potentially beneficial if aligning with physiological norms. However, elevated WSS should also be monitored for potential long-term effects, depending on the vascular context.

The assessment of TAWSS, an integrated hemodynamic parameter, showed no significant difference between pre- and post-surgery conditions. This outcome can be partially attributed to the similar velocity magnitudes observed throughout the cardiac cycle. Although PeakWSS increases significantly post-operatively (indicating localized shear intensification), the lack of significant change in TAWSS (p=0.23) suggests that, over the full cardiac cycle, the net shear exposure remains comparable. This outcome highlights the temporal filtering effect of the TAWSS metric and underscores the importance of analyzing both instantaneous and time-averaged shear stress measures for a comprehensive assessment of hemodynamic changes.

Surgery involving aortic valve repair in the studied cohort indicated a significant reduction in OSI, attributed to the axial velocity profile alternating between positive and negative flow during the cardiac cycle [23,24]. The specific variation in OSI pre-operatively is seen to be linked to a more anteriorly directed jet with a large recirculation zone on the posterior side of the aortic root. In contrast, post-operatively at peak systole, the axial velocity data in the ascending aorta demonstrate a more symmetric top-hat profile, aligning well with previous observations [2], c.f., Figure 3. The steeper velocity gradient resulted in a higher axial shear stress in the shear layer post-operatively. In the study by [27], it was observed that a non-zero OSI combined with significant WSS increases the risk of aortic wall damage, including the progression of atherosclerosis [28,29]. We can assume that these same biomechanical insults may similarly lead to progressive damage of the valve leaflets. Moreover, the observed reductions in OSI may serve as potential predictors for future complications, such as valve deterioration and aortic root dilation. Consequently, incorporating comprehensive flow dynamic analyses into the post-operative evaluation could improve long-term outcomes and guide more precise patient management strategies.

PeakWSS, like OSI, shows a significant difference between pre- and post-operative cases (p=0.0044), suggesting both metrics may serve as effective parameters for perioperative assessment of the aortic valve. Additionally, Figure 6a illustrates that the change in PeakWSS is more pronounced in Group 1 compared to Group 2, indicating a greater capacity for restoring normal hemodynamics and tissue biomechanics through valve repair in dysfunctional valves, compared to valve-sparing root replacement in already well-functioning valves. The observed hemodynamic normalization following VSRR reinforces a mechanical association between altered WSS/OSI and structural remodeling of the aortic valve [18,19].

The quantitative assessment of axial velocity and shear stress offers valuable insights into the relationship between hemodynamics in the proximal thoracic aorta and variations in aortic valve morphology. BSI technology is promising as a potential tool for perioperative assessment of the blood flow through the aortic valve, providing more granular insight into favorable versus unfavorable flow characteristics. Furthermore, in addition to assessing immediate post-operative hemodynamics, BSI may potentially help predict the long-term durability of any valve repairs. This technique provides a more comprehensive understanding of surgical intervention completeness compared to traditional Doppler and color Doppler assessments of pressure gradients or regurgitation. Additionally, it may offer early indications of potential recurrence. Future investigations with larger cohorts and long-term follow-up will be essential to establish the clinical utility of these findings and to assess their abilities to predict outcomes.

## 5. Limitations

The relatively small number of patients presenting with both aortic valve abnormalities and aortic root dilation limited the ability to perform a robust subgroup analysis. This constraint reduced statistical power and made it challenging to draw definitive conclusions about this specific patient subset. Another limitation of this study is the 2D nature of flow analysis. This is important in the aortic root because of the eccentric flow seen in patients with valve abnormalities and dilated aortic roots, which may be unaccounted for.

The velocity component orthogonal to the imaging plane was not available for all patients in the present study. Capturing the maximum velocity can be optimized by aligning it with the long axis of the left ventricular outflow tract and aorta.

In the cohort under study, which is predominantly characterized by trileaflet/trisinuate aortic valve morphology, WSS is expected to be primarily influenced by the long-axis velocity component. However, in two cases with bileaflet morphology, one of which also exhibited bisinuate root anatomy, previous studies using 4D flow MRI [25,30] and fluid–structure-interaction (FSI) simulations [16], have identified a significant short-axis velocity component. This may give rise to tangential WSS due to the presence of helical flow structures. Nonetheless, the magnitude of tangential WSS in bileaflet cases is generally lower than that of the WSS in the long-axis direction. Therefore, we do not expect this to significantly affect the overall results for the cohort. It is important to note, however, that vorticity information, as well as any jet or transitional laminar-turbulent flow deviating outside the 2D imaging plane, may be lost, which could limit the ability to fully capture the complexity of flow patterns in some cases. Analyzing the helical flow structure through velocity components in the short-axis view may indeed provide valuable insight into OSI and valve function, particularly in cases with bileaflet morphology. This aspect is intended to be explored in future work.

Our previous FSI studies focused on stenotic BAV morphologies characterized by fusion between two adjacent leaflets, along with variations in the interleaflet triangle [15] and commissural angle [16]. In the present study, we examine patients with non-stenotic congenital aortic root disease characterized by varying degrees of regurgitation, ranging from mild to severe. Therefore, a direct side-by-side comparison with our earlier FSI studies was not applicable.

The 3D physiological displacement of the heart throughout the cardiac cycle, which evolves both spatially and temporally, is also not fully captured. This influences the flow quantification of relative velocities to the walls in the left ventricle, as it experiences significant volume changes during systolic contraction. However, for vessels that undergo moderate displacement and area changes, this factor is less significant.

## 6. Conclusions

This study highlights the significant impact of aortic valve repair and aortic root remodeling using valve-sparing techniques on key hemodynamic parameters, PeakWSS, TAWSS, and OSI, at the level of the aortic virtual basal ring in children and young adults. The results show statistically significant differences in both PeakWSS and OSI between pre-operative and post-operative states, suggesting that these parameters may serve as useful markers for monitoring the functional outcome of aortic valve interventions. Despite these promising findings, further research is essential to thoroughly investigate and validate the relationship between OSI reduction and long-term outcomes. Continued evaluation will help in understanding the prognostic implications and guide clinical decision-making for this patient population. 

## Figures and Tables

**Figure 1 bioengineering-12-00776-f001:**
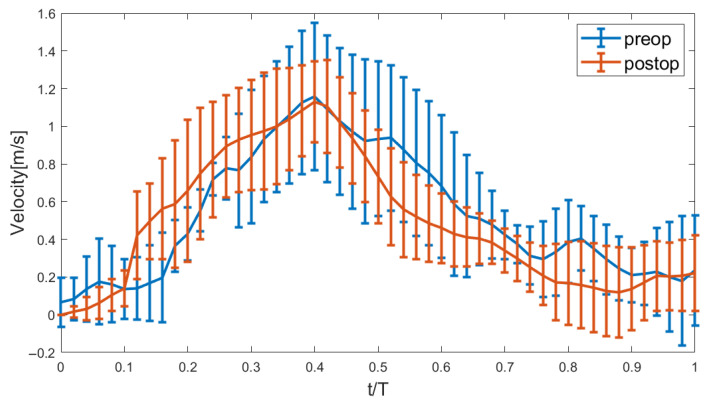
Error bar showing the mean and standard deviation of the peak velocity over the normalized time.

**Figure 2 bioengineering-12-00776-f002:**
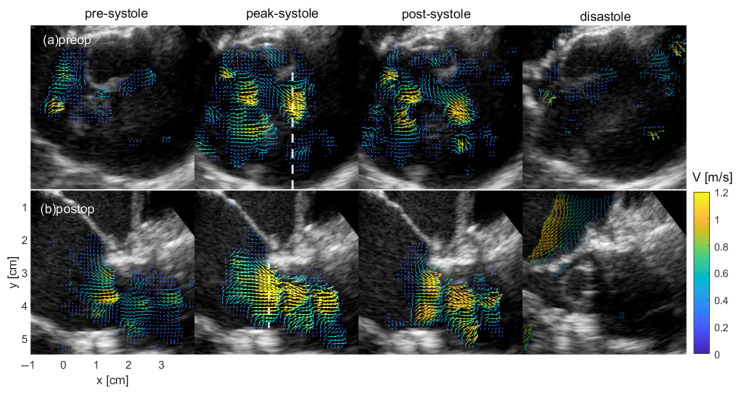
Transesophageal long-axis blood speckle imaging view of the left ventricular (LV) outflow tract and the aortic valve showing the blood flow velocity during different time instants of the cardiac cycle of Patient 1. The velocity vectors are colored with the velocity magnitude. The top of the image is posteriorly positioned towards the left atrium and the bottom is anteriorly positioned toward the right ventricle. (**a**) Pre-operative images are displayed across the top panels. (**b**) Post-operative images are displayed across the bottom panels.

**Figure 3 bioengineering-12-00776-f003:**
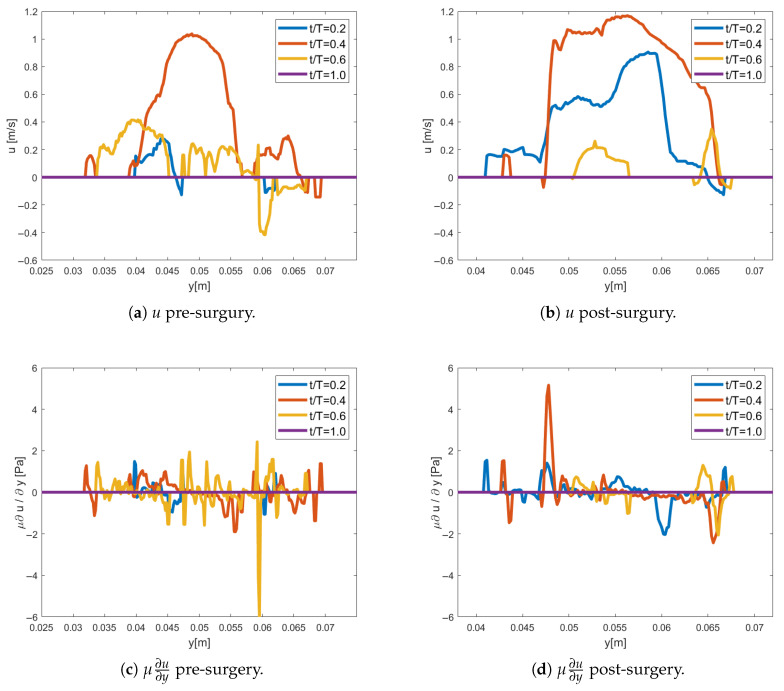
Axial velocity, *u*, and axial shear stress, $\mu \frac{\partial u}{\partial y}$, is shown pre- and post-surgery for Patient 1 at different instances of the cardiac cycle. The profiles are located along the white dashed line annotated in Figure 2. The y-direction corresponds to the short axis, where lower y-values indicate proximity to the anterior wall (higher up on the screen and closer to the TEE probe) and higher y-values indicate proximity to the posterior wall (further from the TEE probe).

**Figure 4 bioengineering-12-00776-f004:**
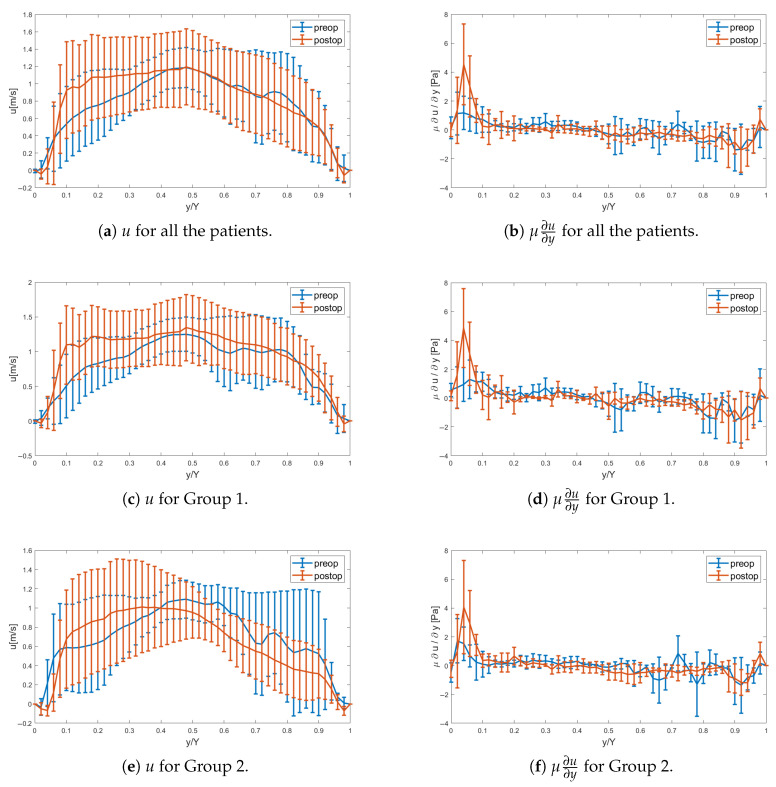
Error bar showing the mean and standard deviation of axial velocity, *u*, and axial shear stress, $\mu \frac{\partial u}{\partial y}$ for Patients 1–10 at peak-systole.

**Figure 5 bioengineering-12-00776-f005:**
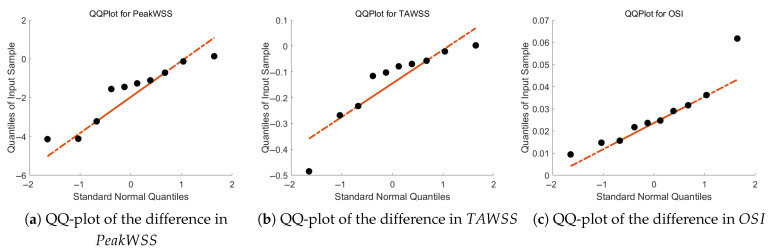
QQ-plot of the difference in PeakWSS, TAWSS, and OSI between pre-surgery and post-surgery.

**Figure 6 bioengineering-12-00776-f006:**
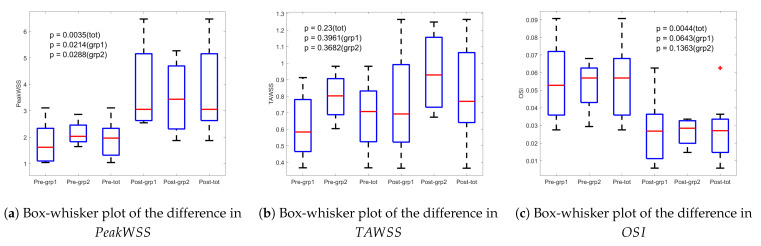
Box-whisker plot of the difference in PeakWSS, TAWSS, and OSI. Which is showing a significant change in OSI following surgery at *p*-value = 0.0044.

**Figure 7 bioengineering-12-00776-f007:**
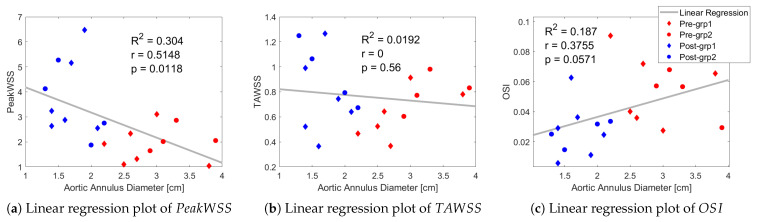
Linear regression showing a correlation between aortic virtual basal ring or “annulus” diameter and PeakWSS, TAWSS, and OSI.

**Table 1 bioengineering-12-00776-t001:** Patient demographics and clinical data.

Patient#	Age(Years)	Gender	Height(cm)	Weight(kg)	BSA(m^2^)	Diagnosis	Surgical Indication	(Neo-)Aortic Valve Morphology	Surgery Performed
Group 1—Dysfunctional aortic valve (>mild stenosis and/or >mild regurgitation)
1	20	Male	177.4	81.4	2.00	LAA isomerism, AVSD s/p repair and aortic valvotomy	severe AR and severe AoR and AAo dilation	Trileaflet/trisinuate with asymmetric leaflets	VSRR and AAoR
2	20	Male	164.5	75.0	1.85	Repaired tetralogy of Fallot	severe AR	Trileaflet/trisinuate with right leaflet bending	Internal annuloplasty with AoV repair
3	8	Male	128.9	31.2	1.06	Repaired perimembranous VSD and AoV	severe AR and severe AoR dilation	Trileaflet/trisinuate	VSRR and AAoR with AoV repair
4	14	Male	170.0	73.1	1.86	Arterial switch operation for TGA	severe AR and severe AoR dilation	Trileaflet/trisinuate	VSRR and AAoR with AoV repair
5	26	Male	170.0	75.3	1.89	Functionally bileaflet AoV	severe AR	Functionally bileaflet/trisinuate with fusion between the coronary leaflets	External annuloplasty and AoV repair maintaining bileaflet configuration
6	25	Female	157.0	102.3	2.11	LAA isomerism, AVSD s/p repair	Moderate AR and severe AAo dilation	Trileaflet/trisinuate with asymmetric leaflets	VSRR and AAoR with AoV repair
Mean ± SD	18.8 ± 6.8		161.3 ± 17.3	73.1 ± 23.2	1.8 ± 0.4				

Group 2—Well-functioning aortic valve (</=mild stenosis and/or </=mild regurgitation)
7	22	Male	190.4	86.6	2.14	Functionally bileaflet AoV	severe AAo dilation	Bileaflet/bisinuate	VSRR and AAo replacement with AoV repair
8	42	Female	160.0	76.2	1.84	Ross-Konno procedure for functionally unileaflet AoV and subaortic stenosis	severe neo-AoR dilation	Trileaflet/trisinuate with asymmetric enlargement of anteriror-right facing sinus	VSRR
9	20	Male	170.0	64.8	1.75	Functionally unileaflet AoV	severe AAo dilation	Functionally unileaflet/trisuate	VSRR and AAo replacement with bileaflet AoV repair including dividing right and non-coronary leaflet commissure
10	13	Female	140.5	40	1.25	Functionally unileaflet aortic valve, aortic coarctation repair, Turner syndrome	severe AAo dilation	Functionally unileaflet/trisunate	AAo replacement with bileaflet AoV repair including dividing right and non-coronary leaflet commissure
Group 2 Mean ± SD	24.3 ± 12.4		165.2 ± 20.8	66.9 ± 20.1	1.7 ± 0.4				
Total Cohort Mean ± SD	21.0 ± 9.2		162.9 ± 17.7	70.6 ± 21.0	1.8 ± 0.4				

AAo, ascending aortic; AAoR, ascending aortic replacement; AR, aortic regurgitation; AoR, aortic root; AoV; aortic valve; AVSD, atrioventricular septal defect; BSA, body surface area; LAA, left atrial appendage; SD, standard deviation; TGA, transposition of the great arteries; VSRR, valve-sparing root replacement; VSD, ventricular septal defect.

**Table 2 bioengineering-12-00776-t002:** Aortic dimensions and aortic valve function.

Patient #	Aortic Root Dilation	Aortic Root Size (cm)	Ascending Aortic Dilation	Ascending Aortic Size (cm)	Pre-op Peak Gradient (mmHg)	Post-op Peak Gradient (mmHg)	Pre-op Aortic Regurgitation	Post-op Aortic Regurgitation
Group 1—Dysfunctional aortic valve (>mild stenosis * and/or >mild regurgitation)
1	Severe	5.2	Severe	5.5	11	9	Severe	Mild
2	Mild	4.2	None	3.3	4	11	Severe	None
3 **	Severe	3.8	Moderate	2.6	8	5	Severe	Mild
4 **	Severe	4.6	Mild	3.2	6	15	Severe	Mild
5	Mild	3.7	None	2.7	15	12	Severe	None
6	Moderate	4.5	Severe	5.0	3	6	Moderate	None
Group 1Mean ± SD	2.2 ± 0.9 ^*o*^	4.3 ± 0.6	1.5 ± 1.4 ^*o*^	3.7 ± 1.2	7.8 ± 4.5	9.7 ± 3.8	2.8 ± 0.4 ^*o*^	0.5 ± 0.5 ^*o*^

Group 2—Well-functioning aortic valve (</=mild stenosis * and </=mild regurgitation)
7	Mild	4.0	Severe	4.8	15	17	Mild	None
8	Severe	5.0	None	3.3	5	12	Mild	None
9	Moderate	4.6	Severe	5.2	12	17	Mild	Mild
10 ***	Mild	3.0	Severe	3.9	6	5	None	Mild
Group 2 Mean ± SD	1.8 ± 1.0 ^*o*^	4.2 ± 0.9	2.3 ± 1.5 ^*o*^	4.3 ± 0.9	12.8 ± 5.7	12.8 ± 5.7	0.8 ± 0.5 ^*o*^	0.5 ± 0.6 ^*o*^
Total Mean ± SD	2.0 ± 0.9 ^*o*^	4.3 ± 0.7	1.8 ± 1.4 ^*o*^	4.0 ± 1.1	10.9 ± 4.6	10.9 ± 4.6	2.0 ± 1.2 ^*o*^	0.5 ± 0.6 ^*o*^

* Mild aortic valvar stenosis is defined as a peak gradient >15 mmHg. ** Diagnosis of aortic dilation for pediatricaged patient is based on Z-scores. *** Diagnosis of aortic dilation for pediatric-aged patient with Turner syndrome is based on Turner Z-scores and aortic size index. ^*o*^ The mean and standard deviation are calculated by substituting: severe by “3”, moderate by “2”, mild by “1”, and none by “0”.

**Table 3 bioengineering-12-00776-t003:** The aortic annulus diameter (AAD), peak wall shear stress (PeakWSS), time average wall shear stress (TAWSS), and oscillatory shear index (OSI) values of ten different patients pre- and post-surgery at y/Y=0.

	AAD [cm]		PeakWSS [Pa]		TAWSS [Pa]		OSI [-]	
**Patient #**	**pre**	**post**	**pre**	**post**	**pre**	**post**	**pre**	**post**
Group 1—Dysfunctional aortic valve								
1	3.8	1.7	1.0439	5.16	0.7804	1.2653	0.0654	0.0364
2	2.2	1.4	1.9215	2.6322	0.4656	0.5226	0.0907	0.029
3	2.7	1.6	1.3221	2.8742	0.3671	0.3647	0.072	0.0626
4	2.5	2.1	1.1023	2.5471	0.5251	0.641	0.0402	0.0246
5	2.6	1.9	2.3378	6.4748	0.6425	0.7451	0.0359	0.0112
6	3.0	1.4	3.1110	3.2391	0.9131	0.9918	0.0275	0.0058
Mean	2.8	1.7	1.8065	3.8212	0.6156	0.7551	0.0553	0.0283
Median	2.7	1.7	1.6218	3.0567	0.5838	0.6930	0.0528	0.0268
SD	0.5	0.3	0.8136	1.6191	0.2044	0.3274	0.0245	0.0203
Group 2—Well-functioning aortic valve								
7	3.9	1.5	2.0554	5.2726	0.8321	1.0644	0.0294	0.0147
8	2.9	2.2	1.6481	2.7535	0.6043	0.6737	0.0572	0.0336
9	3.3	1.3	2.8653	4.1259	0.9816	1.2496	0.0567	0.0251
10	3.1	2.0	2.0153	1.8762	0.7728	0.7938	0.068	0.0318
Mean	3.3	1.8	2.1460	3.5071	0.7977	0.9454	0.0528	0.0263
Median	3.2	1.8	2.0354	3.4397	0.8025	0.9291	0.057	0.0285
SD	0.4	0.4	0.5134	1.4975	0.156	0.2604	0.0165	0.0086
Tot Mean	3.0	1.7	1.9423	3.6956	0.6885	0.8312	0.0543	0.0275
Tot Median	3.0	1.7	1.9684	3.0567	0.7077	0.7695	0.057	0.0271
Tot SD	0.5	0.3	0.6974	1.4934	0.2004	0.303	0.0207	0.0159

## Data Availability

Data available on request due to restrictions in repository access.

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
