# Peer review of "Flow Characteristics by Blood Speckle Imaging in Non-Stenotic Congenital Aortic Root Disease Surrounding Valve-Preserving Operations"

_bioengineering, 2025, doi:10.3390/bioengineering12070776_

Round 1

Reviewer 1 Report

Comments and Suggestions for Authors

The article analyzes blood flow in the aortic root measured using echocardiographic-based blood speckle imaging (BSI) in patients with congenital aortic valve pathology. According to the authors, the new transesophageal echocardiography method allows assessment of not only flow rates but also structural hydrodynamic features of blood flow.

Overall, the article is very interesting and presents new data; however, it does not account for some fundamental characteristics of blood flow, which may reduce the reliability of the observed effects.

The inability to account for blood flow vorticity in 2D scanning does not diminish the merits of the article but should still be mentioned.

  1. It is known that the blood jet ejected into the aorta has a helical structure. This determines the directions of shear stress application on the aortic wall, which vary depending on the ratio of the azimuthal and longitudinal velocity components in the aorta throughout the cardiac cycle.

  2. Flow oscillations and the presence of an intense reverse jet in the aortic lumen are characteristic of normal left ventricular ejection and are not damaging factors. The reverse jet also has a helical structure. Therefore, changes in the OSI (oscillatory shear index) value do not necessarily indicate greater or lesser flow organization, whereas the spatial positioning of the forward and reverse jets is crucial for both the proper functioning of the aortic valve and adequate coronary blood flow

Reviewer 2 Report

Comments and Suggestions for Authors

The study titled “Flow Characteristics by Blood Speckle Imaging in Non-Stenotic Congenital Aortic Root Disease Surrounding Valvar-Preserving Operations” by Shihao Liu et al. is important and innovative. This study explores the use of blood speckle imaging (BSI) during intraoperative transesophageal echocardiography (TEE) to assess flow patterns in 10 patients with non-stenotic congenital aortic root disease undergoing valve-sparing aortic root replacement (VSRR) or aortic valve repair.

Although the study sample is small, the study design is reasonable, the data processing is appropriate, and the results are promising, suggesting potential clinical value of BSI in assessing surgical outcomes. There are some areas where I think the manuscript can be improved.

- I suggest that the principles of Blood Speckle Imaging (BSI) should be explained more clearly.  Because of the novelty and innovative nature of this approach, a more detailed description would be helpful for readers to fully understand its mechanisms and potential clinical value.

- It would be helpful if the manuscript included a more comprehensive review of existing BSI studies in cardiovascular imaging, particularly those focusing on aortic flow. As a result of expanding this section, it will be possible to place the present study into a more comprehensive context, to highlight its contributions to the field, and to clarify its uniqueness.

Reviewer 3 Report

Comments and Suggestions for Authors

Reviewer Comments

This is a manuscript evaluating flow characteristics using Blood Speckle Imaging (BSI) in patients with congenital aortic root disease before and after valvar-preserving operations.

Summary:

This study employed the Blood Speckle Imaging (BSI) technique with transesophageal echocardiography (TEE) to quantitatively assess flow parameters, such as Peak Wall Shear Stress (PeakWSS), Time-Averaged Wall Shear Stress (TAWSS), and Oscillatory Shear Index (OSI), in 10 patients with non-stenotic congenital aortic root disease undergoing valvar repair or valvar-sparing root replacement surgery. The findings suggest that these operations significantly improve hemodynamic patterns, notably reducing OSI and increasing PeakWSS.

Overall Impression:

This is a promising study that utilizes BSI, a relatively new imaging modality, to provide quantitative insights into the hemodynamic impact of aortic valvar-preserving surgery. The manuscript is generally well-written, and the figures (e.g., Figure 2, 3, 4) effectively illustrate the changes in flow patterns and parameters. Although the cohort size is small (n=10), this is an elegant preliminary study that offers valuable initial data on a specific patient population and surgical approach, paving the way for further exploration.

Points for Revision:

To enhance the scientific rigor and clarity of the manuscript for publication, the following points require careful consideration and revision:

  1. Overextension of Conclusions on Long-Term Implications: The manuscript mentions in the Abstract and Conclusion that the observed reduction in OSI "may serve as potential predictors for future complications," such as valvar deterioration and aortic root dilation. While the Discussion section and Limitations appropriately acknowledge the need for future studies with larger cohorts and long-term follow-up to validate these findings and their predictive capability, stating this as a "potential predictor" or implying it "may potentially help predict" in the Abstract and Conclusion, without any long-term outcome data presented in the study itself, is speculative and could be perceived as overstating the immediate findings.
    • Recommendation: Please remove the statements regarding the potential predictive value for long-term outcomes (e.g., valvar deterioration, aortic root dilation, diastolic dysfunction) from the Abstract and Conclusion sections. These points are currently speculative hypotheses based on known biomechanical principles and other literature, not conclusions drawn from the data presented in this study. Please relocate these discussions to the Discussion section, where they can be framed appropriately as potential implications or hypotheses for future research to explore and validate, supported by citations to relevant literature on WSS/OSI and vascular pathology, while clearly stating that this study provides immediate postoperative insights but does not include the long-term follow-up needed for validation.
  2. Abstract Content and Clarity: The Abstract introduces the study's focus on quantitative flow parameters including PeakWSS, TAWSS, and OSI. While it highlights the significant reduction in OSI with specific pre/post values and p-value, it does not present the main findings for PeakWSS and TAWSS in the same detail within the Abstract summary. Since the study measures and analyzes all three parameters extensively, the Abstract should ideally provide a concise summary of the key findings for all primary hemodynamic parameters evaluated (PeakWSS, TAWSS, and OSI) to give readers a complete overview of the main results.
    • Recommendation: Please revise the Abstract to include a brief summary of the statistically significant changes (or lack thereof) observed for PeakWSS and TAWSS, similar to how the OSI results are presented. This will ensure the Abstract accurately reflects the study's key quantitative findings across all measured parameters.
  3. Inconsistency in Total PeakWSS p-value: There is a discrepancy in the reported p-value for the overall difference in PeakWSS between pre- and post-operative states.
    • In the main body of the Discussion and in Figure 6a, the overall comparison shows a significant increase in PeakWSS with a p-value of 0.0035.
    • However, later in the Discussion, when discussing PeakWSS as a parameter for perioperative assessment, it is stated that PeakWSS shows a significant difference with a p-value of 0.0234.
    • These two p-values refer to the same statistical comparison (total cohort pre- vs. post-operative PeakWSS difference) but are numerically different. Please carefully re-check your statistical analysis and ensure consistency by correcting the p-value in all relevant parts of the manuscript and figures.
  4. Limitations in Study Design: As noted by the authors, the study has inherent limitations that affect the interpretation of the findings.
    • Small Sample Size (n=10): The small cohort size significantly limits the statistical power, particularly for subgroup analyses (Group 1 vs. Group 2). While observations on potential differences between groups are presented, definitive conclusions are challenging to draw from such a small sample size.
    • 2D Flow Analysis: The 2D nature of the BSI analysis is a limitation, especially in the aortic root, which can exhibit complex, eccentric, or three-dimensional flow patterns. As acknowledged, this may lead to missing information about velocities orthogonal to the imaging plane, vorticity, or out-of-plane jet/turbulent flow.
  5. Obvious Typos: There are a couple of noticeable typographical errors that need correction:
    • In the sentence describing the correlation between PeakWSS and aortic annulus diameter, "indicating a week negative correlation" should be "indicating a weak negative correlation."
    • In the title of Figure 7, "PeskWSS" should be "PeakWSS".
    • Please conduct a thorough proofread of the entire manuscript to catch and correct any similar errors.

Concluding Remarks:

This study provides valuable preliminary evidence for the utility of BSI in assessing immediate post-operative hemodynamic changes following aortic valvar-preserving surgery. Addressing the points raised above, particularly the appropriate framing of long-term implications, revising the Abstract to include key quantitative findings for all measured parameters, ensuring accuracy and consistency of statistical results, and careful proofreading, will significantly strengthen the manuscript and its suitability for publication in a journal of the mentioned impact factor. I look forward to reviewing the revised version.

Round 2

Reviewer 2 Report

Comments and Suggestions for Authors

This retrospective study evaluates Blood Speckle Imaging (BSI) for assessing pre- and post-operative flow dynamics in 10 patients with non-stenotic congenital aortic root disease undergoing valvar-preserving surgery. Using intraoperative transesophageal echocardiography, the study measures Peak Wall Shear Stress (PeakWSS), Time-Averaged Wall Shear Stress (TAWSS), and Oscillatory Shear Index (OSI). Key findings include significant post-operative increases in PeakWSS (p=0.0035), reductions in OSI (p=0.0044), and no change in TAWSS (p=0.23), suggesting improved hemodynamics.

The revisions have strengthened the paper, but some areas would still benefit from further clarification.

A clearer explanation of the clinical meaning behind the increased Peak WSS and lowered OSI—and whether these changes pose any risk—would improve the manuscript.

It would be helpful to provide a more detailed description of how the OSI value for Patient 4 was calculated.

Author Response

We sincerely thank the reviewers for their valuable suggestions and thorough review of our manuscript. All changes are highlighted in red in the revised manuscript, and our detailed responses are provided below

Reviewer 2:
Comment1:
This retrospective study evaluates Blood Speckle Imaging (BSI) for assessing pre- and post-operative flow dynamics in 10 patients with non-stenotic congenital aortic root disease undergoing valvar-preserving surgery. Using intraoperative transesophageal echocardiography, the study measures Peak Wall Shear Stress (PeakWSS), Time-Averaged Wall Shear Stress (TAWSS), and Oscillatory Shear Index (OSI). Key findings include significant post-operative increases in PeakWSS (p=0.0035), reductions in OSI (p=0.0044), and no change in TAWSS (p=0.23), suggesting improved hemodynamics.

The revisions have strengthened the paper, but some areas would still benefit from further clarification.

A clearer explanation of the clinical meaning behind the increased Peak WSS and lowered OSI—and whether these changes pose any risk—would improve the manuscript.

Response 1:
The Introduction and Discussion sections have been extended to include a clearer explanation of the clinical significance of increased Peak WSS and lowered OSI. The following is added to the Introduction:
“A perioperative echocardiography study by Hayaschi et al. (2021) showed that patients undergoing VSRR surgery had a reduced OSI and increased peak WSS compared to controls, suggesting a reduced leaflet stress and potential de-generation following VSRR [15]. A recent proteomic study in LVAD recipients linked abnormal WSS/OSI patterns to reduced expression of cytoskeletal and junction proteins in aortic leaflets, implicating shear forces in endothelial dysfunction, inflammation, and tissue remodeling [16]. While these findings support a mechanical link between WSS/OSI and structural valvar changes, most evidence is based on limited longitudinal data correlating shear stress to clinical outcomes.”
The following is added to the Discussion:
“The observed hemodynamic normalization following AVRR reinforces a mechanical association between altered WSS/OSI and structural remodeling of the aortic valve [15,16].”

Comment 2:
It would be helpful to provide a more detailed description of how the OSI value for Patient 4 was calculated.

Response 2: We appreciate the reviewer’s suggestion. The paragraph is now revised to provide a clearer explanation. The oscillatory shear index (OSI) was calculated using Equation \ref{eq:OSI}. For each patient, the wall region (anterior or posterior) that contributed most significantly to the OSI, as defined by Equation \ref{eq:OSI}, was identified. The OSI values reported in Table \ref{table:TAWSS_OSI_RRT} were calculated using data from this dominant wall region. For the majority of patients, the anterior wall ($y/Y = 0$) exhibited the highest OSI contribution. However, for Patient 4, the posterior wall ($y/Y = 1$) showed the greatest contribution, and thus this region was used in the OSI calculation for that patient.”

Reviewer 3 Report

Comments and Suggestions for Authors

I appreciate the authors’ extensive and detailed revisions, which have substantially enhanced the manuscript. The only remaining issue is the IRB statement: please specify in the Methods section the name of the reviewing board, the IRB approval number or exemption reference (e.g., expedited or waived review). I have no further comments on other revisions. I wish the manuscript a smooth path to its final decision.

Author Response

We sincerely thank the reviewers for their valuable suggestions and thorough review of our manuscript. All changes are highlighted in red in the revised manuscript, and our detailed responses are provided below

Reviewer 3:
Comment 1:
I appreciate the authors’ extensive and detailed revisions, which have substantially enhanced the manuscript. The only remaining issue is the IRB statement: please specify in the Methods section the name of the reviewing board, the IRB approval number or exemption reference (e.g., expedited or waived review). I have no further comments on other revisions. I wish the manuscript a smooth path to its final decision.
Response 1:
The following IRB statement is now included in the Methods section.
“The Cleveland Clinic Institutional Review Board waived ethical review and approval due to the retrospective nature of the study and the use of de-identified data.”